# Interaction of Pregnancy-Specific Glycoprotein 1 With Integrin α5β1 Is a Modulator of Extravillous Trophoblast Functions

**DOI:** 10.3390/cells8111369

**Published:** 2019-10-31

**Authors:** Shemona Rattila, Caroline E. Dunk, Michelle Im, Olga Grichenko, Yan Zhou, Marie Cohen, Maria Yanez-Mo, Sandra M. Blois, Kenneth M. Yamada, Offer Erez, Nardhy Gomez-Lopez, Stephen J. Lye, Boris Hinz, Roberto Romero, Gabriela Dveksler

**Affiliations:** 1Department of Pathology, Uniformed Services University of Health Sciences, Bethesda, MD 20814, USA; shemona.rattila@usuhs.edu (S.R.); olgri4@gmail.com (O.G.); 2Lunenfeld Tanenbaum Research Institute, Sinai Health System, Toronto, ON M5T 3H7, Canada; dunk@lunenfeld.ca (C.E.D.); Lye@lunenfeld.ca (S.J.L.); 3Laboratory of Tissue Repair and Regeneration, Faculty of Dentistry, University of Toronto, Toronto, ON M5G 1G6, Canada; michelleim27@gmail.com (M.I.); boris.hinz@utoronto.ca (B.H.); 4Department of Obstetrics, Gynecology and Reproductive Sciences, Center for Reproductive Sciences, University of California San Francisco, San Francisco, CA 94143, USA; Yan.Zhou@ucsf.edu; 5Department of Pediatrics, Gynecology and Obstetrics, University of Geneva, 1206 Geneva, Switzerland; Marie.Cohen@hcuge.ch; 6Department of Molecular Biology, Universidad Autónoma de Madrid (UAM), 28049 Madrid, Spain; maria.yanez@cbm.csic.es; 7Experimental and Clinical Research Center, a Cooperation between the Max Delbrück Center for Molecular Medicine in the Helmholtz Association, and the Charité-Universitätsmedizin Berlin, AG GlycoImmunology, 13125 Berlin, Germany; sandra.blois@gmail.com; 8Berlin Institute of Health, Institute for Medical Immunology, Charité-Universitätsmedizin Berlin, corporate member of Freie Universität Berlin, Humboldt-Universität zu Berlin, 13353 Berlin, Germany; 9Cell Biology Section, National Institute of Dental and Craniofacial Research, National Institute of Health, Bethesda, MD 20892, USA; kyamada@dir.nidcr.nih.gov; 10Perinatology Research Branch, Division of Obstetrics and Maternal-Fetal Medicine, Division of Intramural Research, *Eunice Kennedy Shriver* National Institute of Child Health and Human Development, National Institutes of Health, U. S. Department of Health and Human Services, Bethesda, MD 20892, and Detroit, MI 48201, USA; offererez@gmail.com (O.E.); ngomezlo@med.wayne.edu (N.G.-L.); 11Maternity Department “D,” Division of Obstetrics and Gynecology, Soroka University Medical Center, School of Medicine, Faculty of Health Sciences, Ben Gurion University of the Negev, Beer-Sheva 8410501, Israel; 12Department of Obstetrics and Gynecology, Wayne State University School of Medicine, Detroit, MI 48201, USA; 13Department of Biochemistry, Microbiology and Immunology, Wayne State University School of Medicine, Detroit, MI 48201, USA; 14Department of Obstetrics and Gynecology, University of Michigan Health System, Ann Arbor, MI 48109, USA; prbchiefstaff@med.wayne.edu; 15Department of Epidemiology and Biostatistics, Michigan State University, East Lansing, MI 48824, USA; 16Center for Molecular Medicine and Genetics, Wayne State University, Detroit, MI 48201, USA; 17Detroit Medical Center, Detroit, MI 48201, USA; 18Department of Obstetrics and Gynecology, Florida International University, Miami, FL 33199, USA

**Keywords:** pregnancy-specific glycoproteins, extravillous trophoblasts, integrin α5β1, adhesion, migration, preeclampsia

## Abstract

Human pregnancy-specific glycoproteins (PSGs) serve immunomodulatory and pro-angiogenic functions during pregnancy and are mainly expressed by syncytiotrophoblast cells. While PSG mRNA expression in extravillous trophoblasts (EVTs) was reported, the proteins were not previously detected. By immunohistochemistry and immunoblotting, we show that PSGs are expressed by invasive EVTs and co-localize with integrin α5. In addition, we determined that native and recombinant PSG1, the most highly expressed member of the family, binds to α5β1 and induces the formation of focal adhesion structures resulting in adhesion of primary EVTs and EVT-like cell lines under 21% oxygen and 1% oxygen conditions. Furthermore, we found that PSG1 can simultaneously bind to heparan sulfate in the extracellular matrix and to α5β1 on the cell membrane. Wound healing assays and single-cell movement tracking showed that immobilized PSG1 enhances EVT migration. Although PSG1 did not affect EVT invasion in the in vitro assays employed, we found that the serum PSG1 concentration is lower in African-American women diagnosed with early-onset and late-onset preeclampsia, a pregnancy pathology characterized by shallow trophoblast invasion, than in their respective healthy controls only when the fetus was a male; therefore, the reduced expression of this molecule should be considered in the context of preeclampsia as a potential therapy.

## 1. Introduction

Pregnancy-specific glycoproteins (PSGs), also known as Schwangerschafts Protein 1 (SP1) and pregnancy-specific beta 1 glycoproteins, are secreted placental proteins found in the maternal circulation [1,2,3,4]. PSGs belong to the carcinoembryonic antigen cell adhesion molecules (CEACAM) family but while CEACAMs are mostly membrane-associated proteins, PSGs are secreted [5]. All species that express PSGs have multiple PSG-encoding genes [6]. In mice and rats, 17 genes (*Psg16*–*Psg32*) and eight genes (*Psg36–Psg43*) have been identified, respectively [7]. In humans, there are 11 *PSG* genes; while one of them has been reported to be a pseudogene, the expression of 10 PSGs designated as PSG1–PSG9 and PSG11 is predicted, although specific antibodies for each member of the family are not available [8]. The PSG1 messenger RNA (mRNA) is abundantly expressed throughout pregnancy and the protein concentration of PSGs reaches its maximum level in maternal plasma at term [9,10]. PSGs are found only in species with hemochorial placentation in which maternal blood comes into direct contact with fetal cells, posing a risk of rejection by the maternal immune system [11]. Interestingly, our group and others have shown that human PSGs and murine PSG23 have immune-regulatory activity consistent with the hypothesis that these proteins may participate in tolerance to the fetal semi-allograft [12,13,14,15,16]. In addition, we have reported that PSG1 has pro-angiogenic activity as it induces endothelial tubulogenesis [17,18]. Furthermore, human PSG1 and PSG9 and mouse PSG23 were shown to have anti-thrombotic activity [9].

PSGs are expressed predominantly, but not exclusively in trophoblasts as low levels of expression were detected in a healthy colon and in the squamous epithelium of the esophagus [19,20]. PSG expression has also been reported in tumors of trophoblastic and non-trophoblastic origin [21,22,23]. Similar to the presence of human PSGs in non-placental tissues, mouse PSG18 is expressed in the follicle-associated epithelium of Peyer’s patches potentially playing a role in the interplay between epithelial cells and immune cells in mucosa-associated lymphoid tissue [24].

Trophoblasts are a specialized cell population in the placenta serving various functions ranging from attachment, migration and invasion to vascular remodeling [25]. In the human placenta, cytotrophoblasts (CTBs) proliferate and differentiate into spatially distinct populations [26]. In the floating villi, fusion of CTBs generates multinucleated syncytiotrophoblast (STB) [27,28,29]. STB produces pregnancy hormones, transport nutrients and oxygen from the mother to the fetus and remove fetal waste products [27,28]. In the anchoring villi that physically anchor the placenta to the uterine wall, differentiation of CTBs starts with the formation of trophoblast cell-columns, in which the proximal cell-column trophoblasts are highly proliferative, and the distal cell-column trophoblasts are non-proliferative, migratory and eventually differentiate into invasive extravillous trophoblasts (EVTs) [28]. EVTs migrate towards and invade into the maternal decidua to transform the uterine spiral arteries of the fetal-maternal interface [30,31]. In human pregnancy, exclusive expression of PSGs by STB was reported more than two decades ago [32]. More recent studies have indicated the presence of PSG mRNA in EVTs [33,34]. Therefore, we first examined whether PSGs are expressed in EVTs using two PSG-specific antibodies and investigated the interaction of PSG1 with EVTs.

Some PSG1 ligands have been identified; human PSG1 and mouse PSG17, PSG22 and PSG23 bind to heparan sulfate proteoglycans (HSPGs) [17,35,36]. The interaction of PSG1 with HSPGs was shown to be required for the ability of PSG1 to induce endothelial tube formation [17]. Besides binding to HSPGs, PSG1, PSG9 and PSG23 bind to integrin αIIbβ3 and thereby inhibit fibrinogen binding to platelets [9]. As differentiation into EVTs is accompanied by a sequential alteration of integrin expression referred to as “integrin switching”, which is regulated in a spatial and temporal manner, we next investigated the interaction of PSG1 with integrins. Proliferative CTBs anchored to the basement membrane express integrin α6β4 [37]. Their differentiation into EVTs near the distal cell column is accompanied by down-regulation of integrin α6β4 and up-regulation of integrin α5β1 expression, associating with acquisition of a migratory cell phenotype [31,38]. The cells that invade the uterine wall express integrin α1β1 along with α5β1 and loose α6β4 expression [37,39,40]. In this study we show via numerous functional assays that immobilized PSG1 induces adhesion of primary EVTs and EVT-like cell lines in an integrin α5β1-dependent manner and that PSG1 directly interacts with this integrin. Furthermore, EVTs seeded on PSG1 have increased migration when compared to cells seeded on a control protein as determined in wound healing assays and single-cell movement tracking experiments.

The spatial regulation of integrin expression is altered in various placental disorders involving either excessive invasion, such as in placenta accrete, or insufficient invasion, such as in pre-eclampsia (PE), suggesting that alteration of PSG levels may in part be connected to these defects [40,41,42,43]. In this study using a specific ELISA, we report reduced serum concentrations of PSG1 in African American women diagnosed with early-onset and late-onset PE but only when they carried a male fetus. Altogether our results indicate that PSGs play important roles in several processes required for normal placentation.

## 2. Materials and Methods 

### 2.1. Protein Production and Purification

PSG1-Fc, PSG1-His, PSG1N-Fc and the proteins used interchangeably as negative controls, Fc and CEACAM9-Fc, were generated from the supernatant of stably transfected CHO-K1 single-cell clones established in our laboratory and grown in hollow fiber cartridge bioreactors (FiberCell Systems, Frederick, MD, USA), as previously described [44]. Plasmids encoding the complementary DNAs (cDNAs) of the single-domain proteins (PSG1A2-Fc and PSG1B2-Fc) were generated as previously reported [44]. For the generation of the PSG1A2 and PSG1B2-Fc single-domain proteins, the plasmids were transfected into ExpiCHO cells (Thermo Fisher Scientific, Waltham, MA, USA) following the manufacturer’s recommendations and the supernatants were collected 5–6 days post-transfection based on cell viability. PSG1 from the serum of pregnant women and PSG1-His generated from the stable cell line in the bioreactor were purified using an anti-PSG1 mAb#4 column as described previously [16]. The identity of native PSG1 was confirmed by matrix-assisted laser desorption/ionization-time of flight (MALDI-TOF) mass spectrometry (MS). The anti-PSG1 mAb#4, which reacts with the N-terminal domain of PSG1, was obtained from Dr. S. Jonjic (University of Rijeka, Rijeka, Croatia). For the purification of Fc-tagged proteins, the supernatants were applied to protein A columns (GE Healthcare, Chicago, IL, USA) and the proteins were eluted with 0.1 M glycine buffer pH 2.7, followed by immediate neutralization with 1 M Tris-HCl pH 8 (Thermo Fisher Scientific). The eluted fractions were pooled, concentrated, and buffer-exchanged with phosphate-buffered saline (PBS) using an Amicon Ultra-10K centrifugal filter unit (MilliporeSigma, Burlington, MA, USA). For quantitation, the purified proteins were separated on NuPAGE 4%–12% Bis-Tris gels (Life Technologies, Carlsbad, CA, USA) at different dilutions alongside known concentrations of bovine serum albumin (Thermo Fisher Scientific) used as standards. The proteins on the gel were stained with GelCode Blue (Thermo Fisher Scientific) and were quantified by densitometry. The identity of the proteins were confirmed by Western blot with specific Abs.

### 2.2. Cell Lines

The human first trimester trophoblast cell line HTR8/SVneo was provided by Dr. Caroline Dunk (Mount Sinai Hospital, Toronto, Ontario, Canada) and Dr. Charles Graham (Queen’s University, Kingston, Ontario, Canada), and was cultured in RPMI 1640 (Corning, Corning, NY, USA). The human first trimester trophoblast cell line Swan71 was provided by Dr. Gil Mor (Yale University School of Medicine, New Haven, Connecticut, USA) and CHO-K1 cells were obtained from American Type Culture Collection (Manassas, Virginia, USA), and were cultured in Dulbecco’s Modified Eagle Medium (DMEM) (Lonza, Basel, Switzerland). The cell line CHO-B2 (integrin α5 deficient) was a gift from Dr. Rudy L. Juliano (University of North Carolina, Chapel Hill, North Carolina, USA) and was cultured in Minimum Essential Medium (MEM) alpha modified with ribo- and deoxyribonucleosides (Thermo Fisher Scientific). All cell lines were maintained by using the media indicated above, supplemented with 10% fetal bovine serum (FBS) (Innovative Research Inc., Novi, MI, USA), 100 μg/mL penicillin/streptomycin (Corning) and 100 μg/mL normocin (InvivoGen, San Diego, CA, USA) in a 37 °C humidified incubator with 5% CO_2_.

### 2.3. Placental Tissue Collection and Primary Cells Isolation 

Placental and decidual tissues were obtained from women carrying a normal pregnancy after legal termination during the first- or second-trimester of pregnancy. All patients who donated a placenta provided their informed written consent prior to their inclusion in the study in accordance with the Declaration of Helsinki. Approvals on the use of human subject materials were obtained from the following institutions and their respective committees on human research: the Research Ethics Board (REB # 12-0007E) of Mount Sinai Hospital (Toronto, Ontario, Canada), the local ethics committee of the Maternity and Pediatrics Department (project code # Gyn 02-007) of the Geneva University Hospital (Switzerland) and the University of California San Francisco (California, USA). Fresh tissue specimens were collected in ice-cold PBS and were washed several times in sterile Hanks balanced salt solution (HBSS). Placentas were microdissected to collect EVT columns at the tips of the villi. Approximately 30 EVT columns per placenta were digested with 10 mL of an enzyme cocktail (0.125% trypsin, 100 mM HEPES, 40 mM MgSO_4_, 0.15 mg/mL DNase I and 100 µg/mL normocin in HBSS) for 15 min in a shaking water bath at 37 °C. After neutralization with FBS, the cells were resuspended in DMEM and the suspension was filtered through 70 µm mesh and counted. The cells were confirmed as >95% EVTs based on the expression of cytokeratin 8 (CK8), and human leukocyte antigen-G (HLA-G) and the absence of vimentin and then used for single-cell tracking experiments shown in Figure 1D–F. The EVTs utilized for the experiments shown in Figure 4B were purified on Percoll gradients as previously described [45].

CTBs were isolated from the chorionic villi of second-trimester placentas and purified on Percoll gradients according to previously described protocols [46,47]. After initial purification of CTBs, the remaining leukocytes were removed with anti-CD45-coupled magnetic beads. Purified CTBs were cultured in serum-free DMEM, with 2% Nutridoma (Boehringer Mannheim Biochemicals, Indianapolis, Indiana, USA) on a Matrigel-coated surface for 14–20 h for differentiation into EVTs.

### 2.4. Fluorescence Immunohistochemistry for Analysis of PSGs Expression and Localization with Integrin α5 in First-Trimester Placentas 

Tissues collected from the placenta at 5–8 weeks of gestation were fixed in paraformaldehyde (PFA) and processed to 5 μm thick paraffin sections on Superfrost Plus glass slides (VWR, Mississauga, Ontario, Canada). Tissue sections were deparaffinized using xylene and rehydrated with a descending concentration gradient of ethanol. Heat-induced antigen retrieval with sodium citrate buffer (pH 6) was performed to avoid epitope masking. Autofluorescence was minimized with the treatment of 0.1% Sudan Black in 70% ethanol for 1 min. Blocking with a cocktail of 10% normal goat serum, 2% human serum, 2% donkey serum and 2% rabbit serum for 1 h was done to prevent non-specific binding. Tissue sections were incubated overnight at 4 °C with the following primary Abs: 1:250 dilution of anti-PSG mouse mAb-BAP3 (cat#GM-0507; Aldevron, Fargo, North Dakota, USA) or anti-PSG mAb #4, which react with the B2 and N-domain of PSG1 respectively, and anti-cytokeratin 8 guinea pig polyclonal (cat# ab194130; Abcam, Cambridge, UK) at a 1:100 dilution. These Abs were applied at appropriate combination for dual immunofluorescence images shown in Figure 1; anti-PSG mouse mAb-BAP3 and anti-integrin α5 rabbit mAb (cat# ab112183; Abcam) at 1:500 dilution were applied in combination for dual immunofluorescence images shown in Figure A1. Serial sections were incubated with anti-HLA-G (cat#11-499-c100; Exbio, Vestec, Czechia) to identify the EVT in the placental sections for Figure 1. As controls, mouse (cat#X0931, Agilient Technologies, Santa Clara, CA, USA), guinea pig (cat# NBP197036; Novus Biologicals, Centennial, CO, USA) and rabbit IgGs (cat#ab171870, Abcam) instead of specific primary Abs were applied at the same concentrations and combinations as the specific primary Abs. Secondary Abs used for Figure 1 were anti-mouse Alexa Fluor-488 (cat#A-28175; Thermo Fisher Scientific) at 1:200 dilution and anti-guinea pig biotin (cat#106-066-003; Jackson ImmunoResearch Laboratories Inc., West Grove, PA, USA) at 1:1000 dilution and amplification was carried out using Streptavidin Alexa Fluor-546 (Thermo Fisher Scientific). Secondary Abs used for Figure A1 were anti-mouse Alexa Fluor-546 (at 1:300) (cat#A-11030; Thermo Fisher Scientific) in combination with anti-rabbit biotin (at 1:1000) (cat#ab6720; Abcam) and amplification was carried out using Streptavidin Alexa Fluor-488 at 1:1000 dilution (cat#S-32354; Thermo Fisher Scientific). Slides were washed in PBS and mounted in aqueous immune-mount (Thermo Fisher Scientific). Cells were counterstained with the nuclear dye DAPI (5 μg/mL). Images were captured using the Quorum Wave FX spinning disc confocal system comprising a Leica DMI 6000B microscope (Leica Microsystems, Wetzlar, Germany) with a Yokogawa Spinning Head and Image EM Hamamatsu EMCCD camera and Velocity imaging software version 6.30 (Quorum Technologies Inc, Puslinch, Ontario, Canada). To ensure fair comparison across gestation, exposure time and laser intensity was kept constant across all images.

### 2.5. Immunoblot Analysis of PSG1 Expression in Second-Trimester Trophoblasts

Cell lysates were prepared from purified second-trimester CTBs following isolation (0 h) or cultured on Matrigel for different periods of time (14 and 20 h). Ten µg of cell lysate was loaded onto a 4%–12% SDS-PAGE gradient gel and transferred to a polyvinylidene fluoride (PVDF) membrane. Anti-PSG mAb-BAP3 (Aldevron, Fargo, North Dakota, USA) and anti-PSG mAb#4 were used at 1 µg/mL. The Anti-α tubulin mAb (mouse) was utilized to normalize protein loading (clone#B-5-1-2, cat#T5168; Sigma-Aldrich, St. Louis, MO). The signal was developed with horseradish peroxidase conjugated (HRP) anti-mouse secondary Abs (cat#715-0350150; Jackson ImmunoResearch Laboratories Inc.). Images were captured and analyzed with ImageJ software.

### 2.6. Cell Adhesion

Adhesion assays were performed in 96-well polystyrene high protein binding wells (Thermo Fisher Scientific). Wells were coated with 100 μL of the purified proteins diluted in PBS at the indicated concentrations and incubated overnight at 4 °C. Wells were then incubated with 300 μL of 1% bovine serum albumin (BSA)/PBS for 1 h at room temperature to block non-specific binding. After blocking, EVT-like cell lines, removed from tissue culture flasks with Accutase (Innovative Cell Technologies, Inc., San Diego, California, USA), were seeded in their corresponding serum-free media supplemented with 1% BSA-10mM HEPES at a final concentration of 2 × 10^4^ cells/well and incubated for 2 h at 37 °C with 5% CO_2_. When purified primary EVTs were applied, 1 × 10^5^ cells/well were seeded. After incubation, wells were washed three times with PBS to remove unbound cells and the remaining cells were incubated for 2 h with CellTiter 96 AQueous One Solution (MTS; Promega, Madison, WI, USA), following the manufacturer’s recommendation, and absorbance at 490 nm was recorded. When indicated, cells were pre-incubated with 500 μM RGD-peptide (GRGDNP) or RGD-control peptide (GRADSP; Enzo Life Sciences, New York, NY, USA), or with echistatin (MilliporeSigma, Burlington, MA, USA) at 1 ug/mL for 30 min at room temperature or with function-blocking mAbs: mouse anti-human integrin α1β1 (clone#5E8D9, cat# 05-246), mouse anti- human integrin αVβ3 (clone#LM609, cat#MAB1976) and mouse anti-human integrin αVβ5 (clone#P1F6, cat#1961Z; Millipore Sigma); rat anti-human integrin α5 mAb16 and rat anti-human integrin β1 mAb13 [48,49] at the indicated concentrations. Mouse isotype control and rat isotype control Abs were purchased from R&D Systems and BD Biosciences, respectively. To determine the effect of heparin on PSG1-mediated cell adhesion, cells were incubated with 50 µg/mL heparin (Millipore Sigma), which was maintained during cell seeding. Experiments in hypoxic conditions with primary EVTs and EVT-like cell lines were performed using a hypoxic chamber in an incubator regulated to maintain 1% oxygen exposure. Hypoxic-like conditions for the EVT-like cell lines were chemically induced with 100 μM CoCl_2_ added for 4 h prior and kept for the duration of the experiment.

### 2.7. Integrin Binding ELISA

Wells of a 96-well plate were coated overnight at 4 °C with 100 μL of protein diluted in PBS at the concentrations indicated. After washing with tris buffered saline-tween (TBST), the wells were blocked with 300 μL/well blocking buffer (1% BSA in TBS) for 2 h at 37 °C. Recombinant human integrin α5β1 (R&D Systems, Minneapolis, MN, USA) prepared in 1% BSA in 1× TBS with 1 mM MnCl_2_ was added to the wells and incubated for 2 h at 37 °C. Following washes, 1 μg/mL of biotin-conjugated anti-human integrin β1 Ab (cat#BAF1778, R&D Systems) was added to the wells for 2 h at room temperature. The wells were washed again followed by the addition of Streptavidin-HRP and TMB substrate (R&D Systems). Reactions were stopped by addition of 2 N of sulfuric acid and the absorbance at 450 nm was recorded using a GloMax Discover multimode detection instrument (Promega, Madison, WI, USA). When indicated, the integrin was added to the wells together with either 5 μM RGD peptide or the RGD-control peptide. 

### 2.8. FAK Phosphorylation Analysis by Western Blot 

Thirty-five mm poly-D-lysine dishes were coated with 30 μg/mL PSG1-Fc or control protein overnight at 4 °C. Cells maintained overnight in media with 2% FBS, were detached with Accutase and were resuspended in 1% BSA containing serum-free media prior to the addition at 3.5 × 10^5^ cells/dish and incubated 60 min at 37 °C. Dishes were placed on ice, washed with PBS and cells were lysed with 200 μL LDS-sample buffer (Invitrogen, Carlsbad, CA, USA) supplemented with protease and phosphatase inhibitors (Millipore Sigma). Protein concentration was determined with the EZQ Protein Quantitation Kit (Invitrogen) and 50 µg were loaded per lane and separated in a NuPAGE 3–8% tris-acetate gel (Thermo Fisher Scientific) before transferring to a PVDF membrane. The blot was probed sequentially with anti-FAK (cat#3285), anti-pFAK Y397 (clone#D20B1, cat#8556) and anti-α tubulin (clone#11H10, cat#2125; Cell Signaling Technology, Danvers, MA, USA) Abs, followed by HRP-conjugated bovine anti-rabbit IgG (cat#sc-2370, Santa Cruz Biotechnology, Inc., Dallas, TX, USA) and the West Pico Chemiluminescent Substrate (Thermo Scientific). Images were captured using Fujifilm laser imager (LAS 4000) (GE Healthcare, Chicago, IL, USA) and quantitated with the ImageJ software Version 1.52r (NIH, Bethesda, MD, USA). 

### 2.9. Focal Adhesion Structure Analysis by Fluorescence Microscopy

HTR8/SVneo cells were seeded in serum free media onto PSG1-Fc (20 µg/mL) or poly-L-lysine (0.01%)-coated coverslips and allowed to adhere for 2 h at 37 °C. After removal of the media, cells were fixed in 4% PFA for 5 min at room temperature (RT), washed twice with TBS, and permeabilized with 0.5% Triton X-100 for 5 min. Cells were incubated with following primary mouse mAbs overnight at 4 °C: anti-paxillin (cat#610051; BD Biosciences, San Jose, CA, USA) and anti-vinculin (cat#V9131, Millipore Sigma), followed by species-matching secondary Abs coupled to Alexa Fluor-488 (cat#A28175, Thermo Fisher Scientific) for 1 h at room temperature. Co-staining was performed with Alexa-Fluor-555-Phalloidin (cat#A34055; Thermo Fisher Scientific) and mounting with Prolong (Invitrogen, Carlsbad, CA, USA) containing DAPI. Images were obtained with a Leica LSM510 inverted confocal microscope and analyzed with the Leica confocal LAS software version3.8.0 (Leica Microsystems, Wetzlar, Germany).

### 2.10. PSG1-ECM Binding Assay

MRC-5 fibroblast cells were seeded at 2 × 10^5^ cells/dish onto 60-mm dishes and grown for 7 days for extracellular matrix (ECM) deposition. After washing extensively with PBS, 1.5 mL of decellularization solution (20 mM NH_4_OH with 0.5% Triton X-100) was applied four times to remove the cells followed by a second round of extensive washing with PBS. The deposited ECM was incubated with or without the addition of heparin (50 µg/mL) for 3 h at 37 °C. PSG1-Fc (10 µg/mL) for 1 h at room temperature. The ECM was washed with PBS for removal of unbound proteins and then lysed with 40 µL 1% Triton X-100 solution supplemented with a protease inhibitor cocktail. Fifty µg of lysed ECM was loaded onto a 10% SDS-PAGE gel and transferred to a PVDF membrane. The following Abs were used for Western blot analysis: goat anti-human Fc at 1:1000 dilution (cat#H10500, Thermo Fisher Scientific), rabbit anti-fibronectin at 1:2000 dilution (cat#F3648, Millipore Sigma), mouse anti-GAPDH and appropriate secondary Abs (1:1000; LI-COR Biosciences, Lincoln, NE, USA).

### 2.11. Wound Healing Assay

Cell migration was measured in a wound healing assay using a 2-chamber culture-insert with a middle gap (Ibidi GmbH, Munich, Germany). The culture-inserts were placed in each well of a 24-well poly-D-lysine plate previously coated overnight with PSG1-His or control protein. Then, either 6.5 × 10^4^ HTR8/SVneo or 6 × 10^4^ Swan71 in 80 μL of serum-free media (SFM) with 1% BSA was seeded in each of the two chambers of the insert and incubated at 37 °C with 5% CO_2_ for 3 h. Cells that did not attach to the plate were gently washed off and the inserts were removed 6–7 h post-cell seeding. The wells were washed with SFM and the inserts filled with 1.5 mL 1% BSA in SFM. Cell migration into the 500 μm gap was recorded by time-lapse microscopy (Leica AF6000 DMI6000B with built in camera DFC 365FX-566903911) for 20 h. Images were captured every 20 min. Movies were edited using the LAS X software version 3.3.2 (Leica Microsystems, Wetzlar, Germany), analyzed with the “Wound Healing ACAS Image Analysis” (MetaVi Labs Inc., Bottrop, Germany) and expressed as gap closure area (measurement of difference between gap area at t = 0 and t = 20 h) and gap closure speed (measurement of gap area covered per unit time). To mimic hypoxic conditions, cells were pre-treated for 4 h with 100 μM CoCl_2_ in serum-containing media. The wound healing assays were carried out with 100 μM CoCl_2_, in SFM with B-27 Supplement (Thermo Fisher Scientific) or using a hypoxic chamber in an incubator regulated to maintain 1% oxygen exposure. Results shown are mean ± S.D. of seven fields of four wells from one representative of three independent experiments.

### 2.12. Tracking of Single Cell Migration

Wells of a 24-well poly-D-lysine plate were coated with 20 µg/mL PSG1-His or control protein overnight at 4 °C. Wells were blocked with 1% BSA/PBS, and HTR8/SVneo and Swan71 cells were seeded in Opti-MEM (Thermo Fisher Scientific) at a density of 5000 cells/well and allowed to adhere overnight at 37 °C in a humidified 5% CO_2_ incubator. For experiments with primary EVTs, a 48-well plate was used and the cells were seeded with 2% FBS containing Opti-MEM at a density of 2500 cells/well. Imaging was performed using a Zeiss Axiovert (Carl Xeiss Microscopy, Jena, Germany) 135 microscope with a temperature-controlled chamber maintained at 37 °C with 5.0% CO_2_. Images were captured at 10 min intervals for a total of 100 frames and were analyzed with ImageJ using the plug-in MTrackJ, which in the ‘Measure’ option provided output as track length, velocity, and distance to previous point for the individual cells that were tracked. Results are shown as mean ± S.D. of four experiments consisting of *n* = 10 for control protein, and *n* = 22 for PSG1-His in each experiment.

### 2.13. HIF1α Stabilization Analysis by Western Blot

Swan71 and HTR8/SVneo cells were plated in standard growth medium at a density of 2.5 × 10^6^ cells or 2.2 × 10^6^ cells per 100-mm dish, respectively. The following day, cells were washed and treated with 100 µM CoCl_2_ (Millipore Sigma) in 0.1% BSA containing media for 4 h. Cells were lysed with 400 µL ice-cold LDS sample buffer (Invitrogen) containing protease inhibitors (Millipore Sigma). The lysates were centrifuged at 15,000 rpm for 30 min and the protein concentration of the soluble material was determined with the EZQ Protein Quantitation Kit (Invitrogen). Fifty µg of protein were loaded per lane and separated on a NuPAGE 4%–8% Bis-Tris gel (Invitrogen). The proteins were transferred to a PVDF membrane and, after blocking, incubated with mouse anti-HIF1α (clone#241809, cat#MAB1536; R&D Systems) and rabbit anti-α-tubulin (clone#11H10, cat#2125; Cell Signaling), followed by the respective HRP-conjugated secondary antibodies. The signal was developed with the SuperSignal West Pico Chemiluminescence substrate (Thermo Fisher Scientific) and visualized in a Fujifilm laser imager (LAS 4000).

### 2.14. Flow Cytometry

Expression of integrin α5β1 in HTR8/SVneo and Swan71 cells was studied by flow cytometry. For flow cytometry, cells were detached with Accutase, and 1 × 10^6^ cells in 100 µL of Fluorescens Activated Cell Sorting (FACS) analysis buffer (PBS with 2% BSA and 0.05% NaN3) were incubated with 1 µg of anti-human α5β1 mAb (clone#HA5, cat#MAB1999; Millipore Sigma) or mouse isotype control (clone#11711, cat#MAB002; R&D systems) for 1 h on ice, followed by 0.25 µg allophycocyanin (APC) conjugated anti-mouse Ig (BD Biosciences) for 30 min. Excess antibodies were removed by washing with the FACS buffer between steps. Cells were analyzed using the BD LSR II (BD Biosciences). A total of 50,000 events were collected for each treatment using the FACS Diva software (BD Biosciences), and the FlowJo software V10.0.8 (BD Biosciences, San Jose, CA, USA)was used for post-acquisition analysis.

### 2.15. Invasion Assays

Cell invasion was examined using a transwell invasion assay and a Biogel cell invasion assay. For the transwell invasion assay, 24-well format cell culture inserts (BD Biosciences) were used. Insert membranes were coated with Matrigel (Corning) or Geltrex (Thermo Fisher) diluted at 0.1 or 0.2 mg/mL with culture media. PSG1-His or control protein at 60 µg/mL was added to the diluted matrix before coating. Coating was performed by incubating at 37 °C for 2 h followed by drying under laminar air flow for 5 h. After rehydrating the dried matrix-coated insert for 2 h with 100 µL SFM, HTR8/SVneo and Swan71 cells in SFM were plated in the inserts at 8 × 10^4^/insert and 6 × 10^4^/inserts, respectively. Serum containing (10% FBS) media was added into the lower wells and plates were incubated at 37 °C for 24 h. Non-invaded cells from the upper surface of the insert were removed by scrubbing with cotton swabs and the cells on the under surface were fixed with 5% glutaraldehyde (Electron Microscopy Sciences) for 10 min, followed by staining with 2% crystal violet (Sigma Aldrich) for 5 min. For each experiment, cells in 10 randomly chosen fields of each filter were counted manually and the experiment was repeated three times. The collagen I coated, 96-well cell migration Biogel assay plates (Enzo Life Sciences) were used for cell invasion assay according to the manufacturer’s instruction with some modifications. HTR8/SVneo and Swan71 cell suspensions prepared with 2% serum containing media were plated at 2.5 × 10^4^/well and 2.2 × 10^4^/well, respectively. Plates were incubated at 37 °C for 2 h in which the cell-free detection zone in the center of the wells became available after the Biogel dissolved. After gentle washes with SFM, 50 µL of Geltrex at 0.1 or 0.2 mg/mL mixed with 60 µg/mL PSG1-His or control protein was added onto the seeded cells in the wells and incubated for 2 h to allow for gel formation. Some wells were used as reference for pre-invasion conditions and fixed immediately without adding Geltrex. After adding 100 µL of media with 1% FBS to each well with the gelled matrix, plates were further incubated for 24 h. After that time, cells were fixed with 5% glutaraldehyde for 10 min and stained with TRITC-phalloidin (Thermo Scientific) for 40 min followed by DAPI (Thermo Scientific) for 5 min. Images were captured using LAX AF600 software compatible with a Leica AF600 fluorescence imaging system with a built in camera (DFC 365FX-566903911). All images were analyzed using ImageJ software Version 1.52r (NIH, Bethesda, MD, USA) and the number of cells invaded into the detection zones was determined by comparison to the reference wells.

### 2.16. Determination of PSG1 Concentration in Serum of Pregnant Women

Plasma samples were collected at the time of a prenatal visit and all patients provided written informed consent prior to sample collection. Women were enrolled as participants of a study conducted at the Center for Advanced Obstetrical Care and Research of the Perinatology Research Branch, NICHD/NIH/DHHS, the Detroit Medical Center and Wayne State University (Detroit, Michigan, USA). Women with multiple gestations, severe chronic maternal morbidity (i.e., renal insufficiency, congestive heart disease and/or chronic respiratory insufficiency), acute maternal morbidity (i.e., asthma exacerbation requiring systemic steroids and/or active hepatitis) or fetal chromosomal abnormalities and congenital anomalies were excluded from the study. PE was defined as new-onset hypertension that developed after 20 weeks of gestation (systolic or diastolic blood pressure ≥140 mm Hg and/or ≥90 mm Hg, respectively, measured on at least two occasions, 4 hours to 1 week apart) and proteinuria (≥300 mg in a 24 h urine collection, or two random urine specimens obtained 4 h to 1 week apart containing ≥1+ by dipstick or one dipstick demonstrating ≥2+ protein) [50]. Early-onset PE was defined as PE diagnosed and delivery ≤34 weeks of gestation, and late-onset PE was defined as PE delivered and delivery ≥34 weeks of gestation [51].

We determined the concentration of PSG1 with the PSG1 Quantikine ELISA kit following the manufacturer’s recommendations (R&D Systems). Serum samples were tested at 1:100 dilution for the early-onset control and early-onset PE groups (obtained at weeks 20.0–33.3 of gestation) and at 1:200 dilution for the late-onset PE and control (obtained between 37 to 41.7 weeks of gestation). For the few samples in which the values of PSG1 obtained were lower than the lowest standard, samples were diluted 1:50 or 1:13 and retested. All samples, obtained from the study population of African American women, were analyzed in triplicate. Data were analyzed using one-way analysis of variance (ANOVA) followed by pairwise comparisons using Tukey’s adjustment for multiple comparisons. Two-way ANOVA, followed by pairwise tests of simple main effects, was used for comparisons stratified by fetal gender.

## 3. Results

### 3.1. Expression of PSGs in EVTs 

While PSG mRNA expression for different members of the family has been reported in EVTs, previous studies using immunohistochemistry identified the STBs of the placenta as the sole site for PSG expression [32,33]. Therefore, we performed dual immunofluorescence staining of tissues obtained from the first-trimester placental villi and decidua basalis to examine PSGs expression employing more sensitive techniques by using two anti-PSG mAbs. EVTs were identified in the 5-week and 7-week old placentas (1A, 1F), and 6.5-week old decidua (1J) with anti-CK-8, and HLA-G antibodies (1D, 1I, 1M). Expression of PSGs was examined with either anti-PSG BAP3 (1B) or anti-PSG mAb#4 (1G, 1K). Co-staining of 5-week and 7-week-old placentas with anti-PSG and anti-CK8 Abs shows that PSGs are localized to the EVT cell edges. Stronger staining was seen in the more distal fully invasive EVTs (Figure 1C,H). As shown in panel 1L, PSG expression is also observed in the EVTs actively invading the decidua and localizing to points possibly associated with focal adhesions (arrows). Only minimal background was observed with control IgGs (1E). In addition, we observed substantial co-localization of PSGs and integrin α5 in the first-trimester EVTs (Figure A1a). PSG expression was also detected in the EVTs that were differentiated in vitro from second-trimester CTBs (Figure A1b) and in second-trimester EVTs as examined by immunohistochemistry (data not shown). Altogether, these results indicate that besides the high expression by STBs, PSGs are expressed by EVTs in the cell column where they colocalize with integrin α5, and by EVTs invading the decidua. In addition, PSGs are expressed by EVTs differentiated in vitro from CTBs.

### 3.2. Immobilized PSG1 Induces Adhesion of EVTs by Binding to Integrin α5β1 

To investigate whether PSG1 interacts with EVTs and affects their adhesive capacity, Swan71 cells were seeded onto non-tissue culture wells that had been pre-coated with either PSG1-Fc or the Fc-tag as control protein. As shown in the pictures taken 2 h post-seeding, cells adhered to the PSG1-coated wells (Figure 2A, top), but not to the control protein-coated wells (Figure 2A, bottom). Similar results were obtained with the HTR8/SVneo cell line (data not shown). We also determined that this PSG1-mediated effect was observed with PSG1 with a His tag and was dose-dependent with a maximal response observed at a PSG1 concentration of 60 μg/mL. (Figure 2B). The PSG1-mediated cell adhesion was inhibited when the cells were pre-treated with the chelating agent EDTA or with the disintegrin echistatin, indicating a requirement for a divalent-cation dependent receptor and a potential involvement of integrins (data not shown). To confirm the requirement of an integrin receptor for the adhesion of cells to PSG1, the cells were incubated with an RGD or control peptide prior to seeding onto the PSG1-coated wells. As shown in Figure 2C, pre-incubation with the RGD peptide prevented adhesion of the cells to PSG1 while the control peptide did not. Studies were then undertaken to define which of the integrins on EVTs were involved in this interaction. EVTs have been reported to express several integrins including α1β1, α4β1, αVβ3, αVβ5 and α5β1 [52]. We found that anti-αVβ3, anti-αVβ5 and anti-α1β1 neutralizing Abs did not inhibit PSG1-mediated cell adhesion (Figure 2D). On the other hand, the adhesion of Swan71 (Figure 2F, left) and HTR8/SVneo (Figure A1c) cells to immobilized PSG1 were significantly reduced in the presence of neutralizing Abs to the integrin α5 and integrin β1 subunits, in ambient oxygen conditions (21%), CoCl_2_-induced hypoxia-like conditions (Figure A1d) and under low oxygen (1%) conditions (data not shown). Hypoxia more closely reflects the conditions during the first-trimester of pregnancy and has been shown to modulate integrin expression [53,54,55,56]. Importantly, expression of α5β1 on the membrane of these cells was confirmed by flow cytometry (Figure A1e).

Given the limited availability of native PSG1 protein, we performed cell adhesion experiments with PSG1 purified from the serum of pregnant women only in 21% oxygen conditions and demonstrated that the adhesion of HTR8/SVneo cells to native PSG1 was reduced by 45% and 90% in the presence of neutralizing antibodies to integrin α5 and integrin β1, respectively (Figure 2E). To determine whether the interaction between PSG1 and α5β1 was also observed in primary cells, we performed adhesion experiments with primary EVTs maintained in 21% and 1% oxygen. Blocking of integrin α5 and integrin β1 in primary EVTs resulted in 70%–90% reduction of the adhesion mediated by PSG1 (Figure 2F, right). Moreover, we observed that PSG1 induced adhesion of CHO-K1 cells, which express integrin α5, but did not induce adhesion of CHO-B2 cells, which lack expression of this integrin [57] (Figure 2G). Thus, these results indicate that immobilized native and recombinant PSG1 induce cell adhesion in an integrin α5β1-dependent manner in primary and EVT-like cell lines under 21% oxygen and 1% oxygen conditions. To determine whether PSG1 interacts directly with integrin α5β1, we performed an ELISA with purified proteins. Figure 2H–J shows that binding of native and recombinant PSG1 to α5β1 is direct and dose-dependent and can be specifically blocked by the RGD-peptide. Altogether, these results indicate that PSG1 is a new ligand for integrin α5β1.

### 3.3. More Than One Domain of PSG1 Mediates EVT Adhesion

PSG1 is comprised of four Ig-like domains designated as N, A1, A2 and B2 and a short cytoplasmic tail of varying length depending on the splice variant [6]. The PSG1-Fc utilized in our initial studies is a natural splice variant that lacks the A1 domain; therefore, we concluded that this domain is not essential for the ability of PSG1 to bind to α5β1 and induce cell adhesion [6]. We generated recombinant single-domain proteins with an Fc tag and tested them for their ability to induce EVT adhesion. We found that the single-domain proteins composed of either the N- or the B2- domain induced adhesion of HTR8/SVneo while the A2 domain did not (Figure 2K). Similar results were obtained with the Swan71 cell line (data not shown).

### 3.4. PSG1 Induces Focal Adhesion (FA) Structures and Focal Adhesion Kinase (FAK) Phosphorylation 

Ligand binding to integrins initiates the recruitment of adaptor proteins and signaling molecules, followed by cytoskeleton reorganization leading to the assembly of focal adhesion (FA) structures [58]. When EVTs were seeded on PSG1-coated covered slips, we observed the assembly of FA structures as determined by the detection of the adaptor proteins vinculin and paxillin associated with F-actin in HTR8/SVneo cells (Figure 3A). In addition, HTR8/SVneo adhesion to PSG1-coated wells induced phosphorylation of the Tyr-397 of FAK, which was not observed in the poly-D-lysine-coated wells (Figure 3B). Similarly, phosphorylation of FAK at Tyr-397 was detected in Swan71 cells when adhered to PSG1-coated wells but not in the cells adhered to poly-D-lysine-coated wells (data not shown).

### 3.5. PSG1 Binds to HSPGs in the ECM and can Concurrently Bind to Integrin α5β1 on the Cell Surface

We observed PSG1-mediated cell adhesion only when PSG1 was immobilized on a solid surface but not when the protein was added to the media (data not shown). Previous results from our group showed that PSG1 binds to the ECM and to HSPGs [13,17]. To confirm these results and to determine whether the interaction of PSG1 with the ECM is mediated by binding to HSPGs, we tested whether PSG1-Fc could bind to the ECM deposited by a fibroblast cell line in the presence or absence of excess heparin, a highly sulfated form of HSPGs [59,60]. MRC-5 fibroblasts were cultured for 7 days for ECM deposition after which the cells were removed as confirmed by the absence of reactivity with the anti-GAPDH Ab (Figure 3C). We observed that binding of PSG1-Fc to the ECM was inhibited in the presence of heparin (Figure 3C). We also examined whether binding of PSG1 to heparin could interfere with PSG1′s ability to induce adhesion of EVTs. We observed that the presence of added heparin did not interfere with PSG1-Fc-mediated adhesion of HTR8/SVneo cells (Figure 3D). These results strongly suggest that PSG1 can concurrently bind to HSPGs in the ECM and to integrin α5β1 on the cell surface.

### 3.6. PSG1 Increases EVT Migration

Integrin α5β1 expression on EVTs is required for their migratory activity [61,62]. Therefore, we explored whether PSG1 can modulate migration of EVTs. We performed wound healing assays and found that Swan71 cells on the PSG1-His-coated wells migrated faster and covered the wound area almost completely in 20 h in comparison to the cells on the control protein-coated wells (Figure 3E). Cells on the PSG1-coated wells showed a faster gap closure speed and covered a larger area, which was 2–2.5 times over that of cells on the control protein-coated wells both in an oxygen concentration of 21% and in the presence of CoCl_2_ (Figure 3F). We observed similar results when the wound healing assay was performed with HTR8/SVneo cells in 21% or 1% oxygen (data not shown). The observed increase in cell migration was not the result of increased proliferation or viability induced by PSG1 as HTR8/SVneo and Swan71 did not proliferate differently or differ in viability on PSG1 when compared to poly-D-lysine coated wells as determined with the MTS assay (data not shown). We also studied cell migration at the single cell level, which allowed us to investigate the effect of PSG1 on the migration of purified first-trimester EVTs (Figure 3G). We observed higher migration of primary EVTs, HTR8/SVneo and Swan71 on the PSG1-coated wells when compared to the control protein-coated wells as determined by measurements of track length, velocity and distance from previous point (D2P) of the tracked cells (Figure 3G). The average track length obtained for primary EVTs, HTR8/SVneo and Swan71 cells on the PSG1-coated wells were 908–1292 µm, which were more than twice of that of the cells on the control protein-coated wells (394–498 µm; 3G, left). The velocities of single cells on the PSG1-coated wells were 0.019–0.033 µm/sec, which were also more than two times higher than the values obtained for the cells on the control protein-coated wells (0.0083–0.01 µm/s; Figure 3G, middle). Similarly, D2P values of single cells on the PSG1-coated wells (19.6–39.4 µm) were almost three times higher than the values obtained for the cells on the wells coated with control protein (8.9–12.4 µm; Figure 3G, right). Altogether, the results indicate that PSG1 induces increased migration of EVTs. 

### 3.7. The PSG1 Concentration Is Significantly Reduced in Pregnant Women Diagnosed with Early-Onset and Late-Onset PE

We explored whether PSG1 can modulate the invasive capacity of HTR8/SVneo and Swan71 cells on Matrigel in transwell and Biogel cell invasion assays. We were unable to detect any significant differences in the invasive capacity of these cells in the presence of PSG1 (data not shown). Invasion of EVTs into the endometrium is an essential process in placenta formation, and shallow invasion of the decidua and incomplete transformation of the spiral arteries by EVTs is believed to be an initiating cause of PE [28,63,64,65,66]. Although we did not observe any direct effect of PSG1 on EVT invasion, these in vitro studies have limitations [67]. Previous reports suggest that lower than normal levels of PSGs are found in some placental pathologies, including PE [68]; therefore, we investigated the PSG1 levels in PE with a validated specific PSG1 ELISA. We measured PSG1 in serum samples obtained from African American women diagnosed with early-onset and late-onset PE and the respective gestational age-matched controls with no pregnancy complications. We found that the concentration of PSG1 was significantly lower in women diagnosed with early-onset and late-onset PE compared to their respective controls (Figure 4A). The data was then analyzed taking into consideration the gender of the fetus. Interestingly, we found no statistical difference in the PSG1 concentration between early-onset or late-onset PE and the respective controls when the fetuses were female. On the other hand, when the fetuses were male, there was a significant difference for early-onset (*p* = 0.002) and late-onset (*p* = 0.001) PE and their respective gestational age-matched controls (Figure 4B).

## 4. Discussion

PSGs were previously shown to be expressed exclusively by STBs by immunohistochemistry; however, some recent studies have reported expression of PSG mRNAs in EVTs [32,33,34,69]. Therefore, to determine whether PSGs are present in EVTs at the protein level, we utilized two specific anti-PSG mAbs that do not react with any member of the CEACAM family, previously shown to be expressed in the placenta (data not shown) [70,71]. At present, there are no available Abs to a specific PSG family member that could be employed for immunohistochemistry. Characterization of the two mAbs utilized in this study showed that mAb#4 binds to the N-terminal domain of PSG1, PSG6, PSG7 and PSG8, and BAP3 binds to the B2 domain of PSG1, PSG3, PSG4, PSG6, PSG7 and PSG8 (data not shown). We observed positive reactivity with both anti-PSG mAbs in EVTs and STB of 5-week and 7-week-old placentas, with more intense staining in STBs as anticipated based on previously published RNA-seq data [34]. Analysis of staining in EVTs showed stronger staining in the more distal areas, suggesting increased expression of PSGs in the EVTs with an increased invasive cell phenotype. PSG expression in the EVTs invading into the decidua was mostly seen as localized puncta, possibly associated with FA structures. The discrepancy in our results from those of Zhou and co-workers, using the BAP3 mAb, is likely due to the difference in the tissue processing; as for these studies, we utilized antigen retrieval [32]. In addition, we showed co-localized expression of PSGs and integrin α5 in the first-trimester EVTs. EVTs differentiated from second-trimester CTBs also showed expression of PSGs as determined by the immunoblot analysis [46,47]. Although our study reports for the first time the presence of human PSGs in invasive EVTs, expression of mouse PSGs has been reported in trophoblast giant cells, which are similar to human EVTs, as well as in spongiotrophoblasts [35,72,73].

Here we show that immobilized PSG1 induces adhesion of two EVT-like cell lines, HTR8/SVneo and Swan71, and primary EVTs. Inhibition of PSG1-mediated adhesion by EDTA, an RGD-containing peptide, or the disintegrin echistatin pointed to the involvement of an integrin receptor [74,75]. Among the RGD-binding integrins, αVβ3, αVβ5 and α5β1 are expressed on trophoblast cells at various stages of differentiation [52]. Integrin switching from α6β4 in villous CTB, to α5β1 in cell-column trophoblasts, and to α1β1 in mature invasive EVTs has been reported in normal placental development [37]. Using function-blocking Abs to different integrins, we determined that PSG1-mediated adhesion was dependent on integrin α5β1, a major fibronectin receptor [76]. The first-trimester placenta grows in a low oxygen or environment required for successful placenta development [77,78]. Since the differentiation of EVTs occurs most extensively in the first-trimester, the developing EVTs in the growing cell-columns of anchoring villi arise within an oxygen gradient with a shift from proliferative to non-proliferative phenotypes and changes in integrin expression, including that of integrin α5 [26]. Integrin α5β1 has been shown to be up-regulated in low oxygen conditions via HIF1α stabilization, which is only expressed in EVTs of the early placenta [28,56]. We observed PSG1-mediated adhesion of EVTs in ambient (21%) and in low (1%) oxygen conditions. In both cases, cell adhesion was dependent on the interaction of PSG1 with α5β1. The requirement for α5β1 integrin expression was further confirmed using a CHO cell line lacking integrin α5 as these cells did not attach to PSG1-coated wells while wild type cells did. Furthermore, we determined that the interaction of α5β1 and native or recombinant PSG1 is direct by ELISA using purified proteins.

An interaction of PSGs with a different integrin, platelet αIIbβ3, has been previously reported by Shanley and co-workers [9]. They observed that PSG1, PSG9 and mouse PSG23 bind to αIIbβ3 and inhibit the interaction of platelets with fibrinogen. Similar to their observation, we show that more than one domain of the protein mediates the interaction with α5β1 integrin as we observed that both the N and B2 domains of PSG1 individually induce EVT adhesion.

PSG1 induced adhesion of EVTs only when immobilized on a well; therefore, as PSG1 was previously shown to bind to HSPGs, we hypothesized that PSG1 may bind simultaneously to the ECM and to α5β1 on the cell membrane [17]. We observed that PSG1 binds to the ECM deposited by fibroblast and that this binding can be inhibited in the presence of excess heparin. On the other hand, PSG1-induced cell adhesion was not inhibited by heparin indicating that cells can potentially bind to PSG1 anchored to the decidual ECM via its interaction with HSPGs, facilitating the attachment of EVTs to the decidua.

Clustering of ligand-engaged integrins leads to FA assembly that involves the recruitment of signaling molecules similar to FAK and cytoskeletal adaptor proteins, such as vinculin and paxillin, to the integrin cytoplasmic tails [79]. FAK, a mediator of integrin-mediated signaling pathways, has been shown to regulate anchorage-dependent behaviors such as cell proliferation, migration and anoikis suppression [79]. The observed localization of vinculin and paxillin only in the EVTs bound to PSG1 and not to poly-D-lysine is consistent with the PSG1-induced FAK phosphorylation [80]. Integrin α5β1-dependent anchorage of trophoblast to the uterine ECM and FAK signaling are well documented as important regulators of EVT migration during early placental development [31,80]. Integrin α5β1 expression in EVTs has been shown to be essential for their migration and IGF-stimulated enhancement in EVT migration is α5β1 dependent [31,61,62]. In accordance with these previous findings, we show that adhesion to PSG1 increased the migratory capacity of primary EVTs and EVT-like cell lines.

The influence of integrin α5 on EVT invasion is controversial with some groups proposing that binding to fibronectin via this integrin has invasion-restraining and others proposing an invasion-promoting effect [37,81]. In our studies using two different invasion assays, we did not observe differences in invasion between cells incubated with PSG1 and control protein. While the interaction of α5β1 with fibronectin likely differs from the interaction of α5β1 with PSG1, the inherent limitations of the in vitro assays employed does not properly address the complexity of the in vivo EVT invasion process including the composition of the ECM and the presence of PSG1 in the matrix rather than in the solution.

Several studies conducted to investigate the correlation of serum PSG levels with various pregnancy complications reported conflicting results [68,82]. However, inherent limitations of the assays used to measure PSG concentration as well as significant inter- and intra- patient variability may explain some of these discrepancies [83,84,85]. Earlier studies reporting on the association between the PSG level and pregnancy complications did not take into consideration the difference in incidence of PE across different ethnicities and have generally involved small numbers of patients insufficient to produce reproducible and generalizable conclusions [86,87,88]. However, some recent studies identified some risk factors that are differentially associated with certain forms of PE, such as African American race for early-onset PE [89]. Recent reports have revisited the association of PE with PSG levels. A proteomic study identified some PSGs as potential predictive markers of early-onset PE [90]. Another study using global RNA profiling has reported down-regulated PSG expression in patients with severe PE [33]. Down-regulation of mRNA coding for some PSG family members and of PSG-derived peptides, had yet to be validated at the protein level [33,90,91]. We measured the concentration of PSG1, the most highly expressed member of the family based on mRNA expression data, to determine whether prior observations that PSGs are found at lower concentrations in pregnant women suffering from PE could be replicated with an ELISA using a validated anti-PSG1 antibody pair [9]. We observed significant differences in PSG1 concentration in African American women diagnosed with early-onset and late-onset PE compared to their gestational age-matched controls. It is likely that a major fraction of PSG1 measured in maternal serum is derived from the multinucleated STB layer of the placenta. Therefore, both the STBs and EVTs can contribute to the circulating levels of PSG1 in the mother. Abnormally low levels of PSG1 in women with preeclampsia may reflect dysfunctional or stressed STBs, which may in turn contribute to the pathogenesis of this placental syndrome [92]. In contrast, in this study we show that staining of PSG in EVT is located at the membrane where its functional interaction with integrin α5 and focal adhesions mediate migration. Due to the limited availability of placental bed biopsies from preeclamptic women we were unable to assess levels of PSG1 in this EVT context in this study. Whether PSGs also affect the initial penetration of the uterine epithelium by blastocytic STBs that occurs in the earliest stages of implantation requires further investigation. Fetal gender-specific differences in gene expression during pregnancy have been reported for cytokines, pro-angiogenic factors, and some galectins [93,94]. In addition, carrying of a male fetus has been identified as an increased risk factor for adverse pregnancy outcomes [95,96]. Therefore, we also analyzed the expression of PSG1 considering a fetal gender. Interestingly, our results indicate that samples obtained from women carrying a male fetus are the major contributors to the observed differences in PSG1 concentration. At present, the mechanisms that regulate the expression of PSGs in the placenta are not completely understood, and whether fetal sex influences their expression has never been investigated. Additional studies using well-validated reagents, which do not cross-react with members of the CEACAM family and can detect only PSG family members, are required to determine whether the concentration of PSGs as a whole are diminished in PE and are also lower than normal in women of different ethnicities.

## Figures and Tables

**Figure 1 cells-08-01369-f001:**
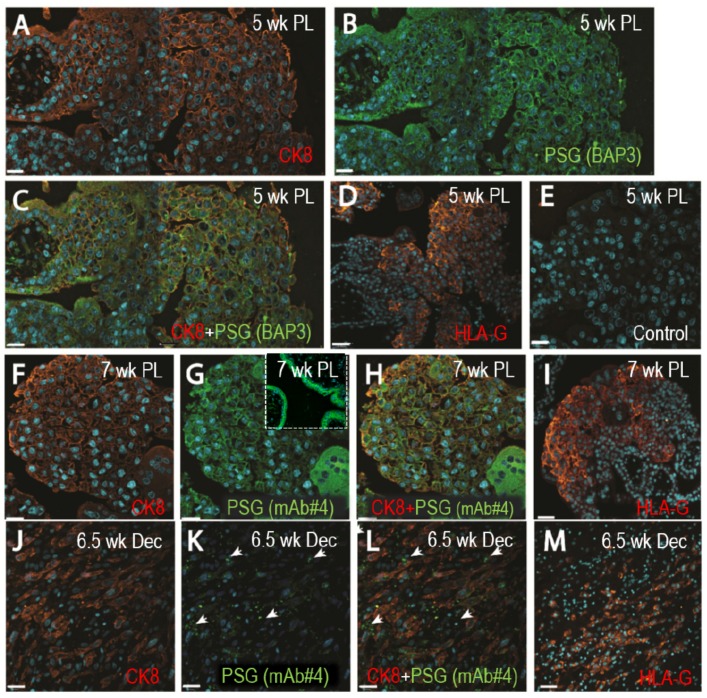
Pregnancy-specific glycoproteins (PSGs) localize to extravillous trophoblasts (EVTs) in first-trimester placenta and interstitial EVTs in the decidua. Representative photographs of dual fluorescent immunohistochemistry localizing cytokeratin 8 (red) and PSGs (green with both BAP3 and #4 mAbs) are shown in both individual and dual channels in sections of first-trimester placenta and decidua basalis. Serial sections stained with a mAb against HLA-G are shown in (**D**), (**I**) and (**M**) (red) to confirm localization in the EVTs. (**A**–**E**) 5-week old placenta. **A**: CK8, **B**: PSG (BAP3), **C**: CK8 and PSG (BAP3), **D**: HLA-G and **E**: guinea pig and mouse IgG control. (**F**–**I**) 7-week old placenta. **F**: CK8, **G**: PSG (mAb#4) with the insert showing the STB layer, **H**: CK8 and PSG (mAb#4) and **I**: HLA-G. (**J**–**M**) 6.5-week old decidua. **J**: CK8, **K:** PSG (mAb#4), **L:** CK8 and PSG (mAb#4) and **M:** HLA-G. Scale bars = 25 µm for (**A**–**C**), (**E**–**H**) and (**J**–**K**), and 45 µm for (**D**), (**I**) and (**M**).

**Figure 2 cells-08-01369-f002:**
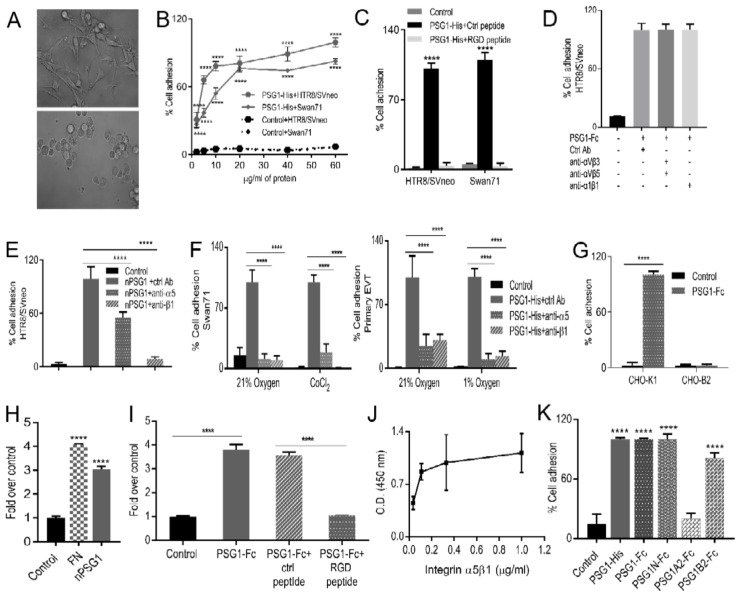
PSG1-mediated adhesion of the EVTs is dependent on a direct interaction with integrin α5β1. Micrographs of Swan71 cells on wells coated with 30 μg/mL PSG1-Fc for 1.5 h in serum-free media (**A**, top) or a protein consisting of just the Fc-tag used as control (**B**, bottom). Images were taken at 40× magnification. (**B**) HTR8/SVneo and Swan71 cells were seeded in wells coated with PSG1-His or control protein at various concentrations. Cells were incubated for 2 h at 37 °C and the wells were washed to remove non-adherent cells. Cells remaining in the wells were incubated with cell titer aqueous solution and cell adhesion was quantified as described in Materials and Methods. The adhesion of Swan71 cells to wells coated with 60 µg/mL PSG1-His is considered as 100%. (**C**) HTR8/SVneo and Swan71 cells, pre-incubated with 500 μM RGD peptide or control (ctrl) peptide for 30 min at room temperature were seeded in wells coated with 30 μg/mL PSG1-His or control protein. The adhesion of control peptide-treated cells to PSG1-His is considered as 100%. (**D**) HTR8/SVneo cells were pre-incubated with the indicated mAbs for 30 min at RT, after which they were seeded in wells coated with 30 μg/mL PSG1-Fc or control protein. (**E**) HTR8/SVneo cells were pre-incubated with the indicated mAbs for 30 min at RT and seeded on wells coated with 30 μg/mL of native PSG1 (nPSG1) or control protein. (**F**) Swan71 cells and purified primary EVTs, pre-incubated with the indicated mAbs were seeded on wells coated with 30 μg/mL PSG1-His or control protein and adhesion assays were performed in 21% oxygen and CoCl_2_-induced hypoxic-like condition or in 1% oxygen conditions as described in Materials and Methods. The adhesion of control Ab-treated cells to PSG1 is considered as 100% in (**D**), (**E**) and (**F**). (**G**) CHO-K1 (integrin α5-expressing) or CHO-B2 (integrin α5-deficient) cells were seeded on wells coated with 30 μg/mL PSG1-Fc or control protein. The adhesion of CHO-K1 cells to PSG1-Fc is considered as 100%. (**H**) nPSG1 or protein control at 20 μg/mL or bovine fibronectin (FN) at 2 μg/mL was coated on wells. After blocking, 1 μg/mL of α5β1 in 1× TBS/1mM MnCl_2_ was added and binding of the integrin was detected with biotin-labeled anti-β1 mAb followed by Streptavidin-HRP. **I.** Wells were coated with PSG1-Fc or protein control (20 μg/mL). After blocking, 1 μg/mL of α5β1 in 1× TBS/1 mM MnCl_2_ was added in combination with 0.5 µM RGD or control peptides and binding of the integrin was detected as indicated in (**H**). The binding of the integrin to the control protein is considered as 1 in (**H**) and (**I**). (**J**) Wells were coated with 20 μg/mL PSG1-Fc or CEACAM9-Fc. After blocking, α5β1 in 1× TBS/1 mM MnCl_2_ was added at various concentrations and binding of the integrin was detected as described above. The graph shows values for PSG1 obtained after subtracting the average of the control values. (**K**). HTR8/SVneo cells were seeded on wells coated with the indicated proteins (20 μg/mL) and adhesion experiment was carried out as described in Figure 2. Cell adhesion to PSG1-His is considered as 100%. Results shown are mean ± S.D. of triplicates from one representative of three independent experiments. *p* values were obtained by a one-way ANOVA followed by Sidak’s multiple comparison tests for (**H**), (**I**) and (**K**), by a two-way ANOVA followed by Tukey’s multiple comparison tests for (**B**), (**C**), (**E**) and (**F**), or followed by Sidak’s multiple comparison tests for (**G**) (**** *p* < 0.0001).

**Figure 3 cells-08-01369-f003:**
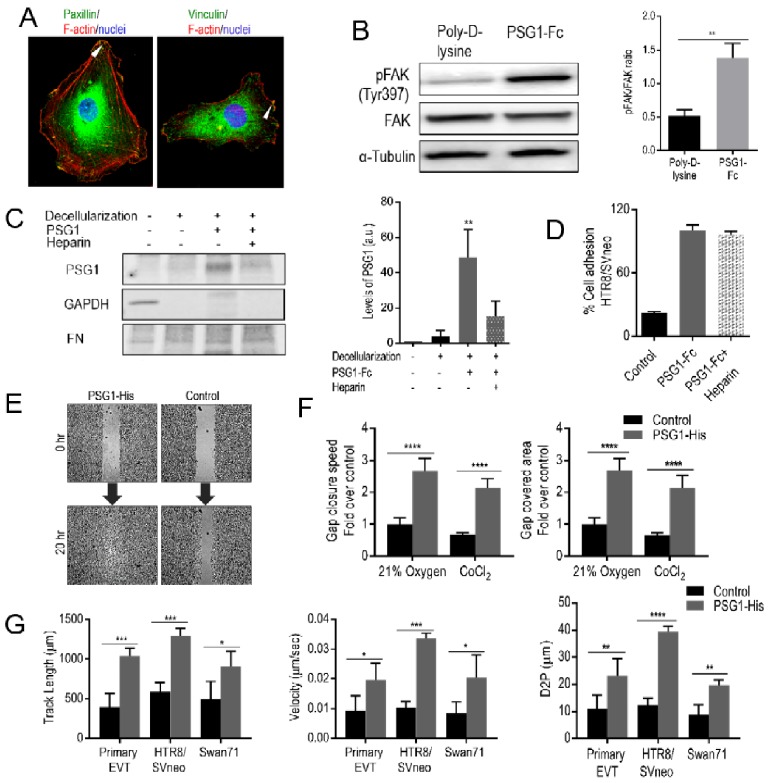
Binding of PSG1 leads to focal adhesion formation and PSG1 can concurrently bind to heparan sulfate proteoglycans (HSPGs) affecting EVT migration. (**A**) Representative immunofluorescence images of HTR8/SVneo cells on PSG1-Fc coated cover slips co-stained with Abs to vinculin, paxillin, and F-actin. DAPI was used to stain the nuclei. Arrowheads show the presence of paxillin (left) and vinculin (right) in the FA structures. Scale bars = 25 µm. (**B**) Representative immunoblot (left) of lysates of HTR8/SVneo cells seeded on poly-D-lysine or PSG1-Fc-coated wells with the corresponding densitometric analysis (right) normalized to total FAK are shown. (**C**) Fibroblast-deposited ECM was decellularized and PSG1-Fc (10 µg/mL) was added in the presence or absence of heparin for 1 h at RT. After several washes, the ECM was lysed and 5 µg was loaded on an SDS-PAGE gel and probed with Abs to human Fc (for PSG1-Fc detection), FN (for ECM detection) and GAPDH (to demonstrate successful cell removal; left). Densitometric analyses of the results are presented in arbitrary units (a.u.; right). (**D**) HTR8/SVneo cells were seeded in wells coated with PSG1-Fc or protein control (30 µg/mL) in the presence or absence of 50 µg/mL heparin. The cell adhesion experiment was carried out as described in Figure 2. The adhesion of untreated cells to PSG1-Fc is considered as 100%. (**E**) PSG1-His or protein control were immobilized on poly-D-lysine-coated wells. Swan71 cells were seeded inside the 2-chamber inserts placed on the protein-coated wells and grown until confluent. Inserts were removed (0 h) to create a cell-free gap area. Cell migration was recorded periodically for 20 h and representative images are shown. (**F**) Quantitative analyses of a representative wound healing assay performed in 21% O_2_ and in CoCl_2-_induced hypoxia-like conditions with Swan71 cells seeded on PSG1-His or control protein-coated wells. Gap closure speed and gap covered area were calculated as described in Materials and Methods. The average of the replicates obtained with the cells seeded on the control protein-coated wells in 21% O_2_ is considered as 1. Results shown are mean ± S.D. of triplicates in (**B**–**D**). (**G**) Quantitative analyses of migration of single cells (primary EVTs, HTR8/SVneo and Swan71) seeded on PSG1-His or control protein-coated wells measured as track length in µm (**G**, left), velocity in µm/sec (**G**, middle) and distance to previous point (D2P) in µm (**G**, right). *p* values were obtained by Student’s *t*-test for **B** (** *p* < 0.003), by a one-way ANOVA followed by Sidak’s multiple comparison tests for **C** (** *p* < 0.002) and by a two-way ANOVA followed by Sidak’s multiple comparison tests for (**F**–**G**) (**** *p* < 0.0001, *** *p* < 0.0002, ** *p* < 0.002 and * *p* < 0.03).

**Figure 4 cells-08-01369-f004:**
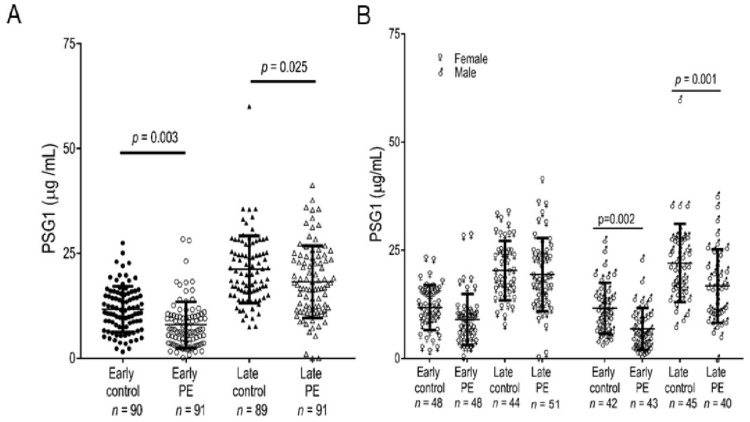
Serum PSG1 concentration is lower in women diagnosed with early-onset and late-onset pre-eclampsia (PE) carrying a male fetus compared to gestational age-matched controls. (**A**) The PSG1 concentration was determined in triplicate in each serum sample using the Quantikine PSG1 ELISA kit. (**B**) PSG1 concentrations shown in A were analyzed considering the gender of the baby. *p* values were obtained by one-way ANOVA followed by pairwise comparisons using Tukey’s adjustment for **A** and by a two-way ANOVA followed by pairwise comparison using Holm-Sidak’s adjustment for **B**.

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
