# Peer review of "Interaction of Pregnancy-Specific Glycoprotein 1 With Integrin α5β1 Is a Modulator of Extravillous Trophoblast Functions"

_cells, 2019, doi:10.3390/cells8111369_

Round 1

Reviewer 1 Report

PSGs represent a family of proteins that are major products of trophoblast cells; however, their biology remains uncertain.  In this report, the authors demonstrate that PSGs are expressed in extravillous trophoblast (EVT) and then go on to comprehensively investigate the role of PSGs in trophoblast cell adhesion, migration, and invasion.  The experimental design is sound and for the most part the interpretation of the results are justified.  The report adds significantly to  our understanding of the biology of PSGs.  Some minor concerns are elaborated below.

1.  There is some concern regarding the use of HTR8 and Swan71 cell lines for the experimentation.  They represent immortalized cell lines with purported features of extravillous trophoblast; however, they are not equivalent to extravillous trophoblast (EVT).   At times in the text the cell lines are equated with EVT, which is not appropriate.  It might be best to refer to these cell lines as EVT-like cells (see below).

Furthermore, investigating invasive properties of an immortalized cell line are potentially problematic.  Are the investigators studying the EVT features of the cells or the features of the cells associated with their immortalization/ transformation? 

In all critical experiments, the authors have supported their observations using the cell lines with experiments with authentic primary EVT cells, which is essential.

2.  CoCl2 and hypoxia are not equivalent.  These conditions share features but responses to CoCl2 are not necessarily the equivalent of responses to low oxygen. 

3.  It would be helpful if the authors provided catalog numbers for the antibodies.

4.  Figures - the quality of the images in the manuscript I received was not optimal.  Hopefully, the quality of figures in the published manuscript will be better.

5.  The authors should be careful regarding their use of terms such as normoxia and hypoxia as applied to cell culture.  Ambient oxygen (~20-21% O2) in an incubator is not normoxia and is not equivalent to the oxygen tension of a cell developing in the placenta/uterus.  Hypoxia is a state defined best by the cellular response to low oxygen.  It might be more advisable to refer to the states as ambient oxygen and low oxygen rather than normoxia and hypoxia.

6.  Is the integrin composition of primary EVT cells versus HTR8 or Swan71 cells the same?

7.  Lines 623 and 624, in this sentence HTR8 and Swan71 cell lines are referred to as EVT cell lines.  This is likely not appropriate.  It may be more appropriate to refer to the cell lines as EVT-like cells, as suggested above.  The authors should take a look at the following report:

Lee CQ, Gardner L, Turco M, Zhao N, Murray MJ, Coleman N, Rossant J, Hemberger M, Moffett A. What Is Trophoblast? A Combination of Criteria Define Human First-Trimester Trophoblast. Stem Cell Reports. 2016 Feb 9;6(2):257-72.

8. Are all of the co-authors contributions stated? Stephen Lye?

Author Response

Point 1: There is some concern regarding the use of HTR8 and Swan71 cell lines for the experimentation.  They represent immortalized cell lines with purported features of extravillous trophoblast; however, they are not equivalent to extravillous trophoblast (EVT).   At times in the text the cell lines are equated with EVT, which is not appropriate.  It might be best to refer to these cell lines as EVT-like cells (see below).

Furthermore, investigating invasive properties of an immortalized cell line are potentially problematic.  Are the investigators studying the EVT features of the cells or the features of the cells associated with their immortalization/ transformation? 

In all critical experiments, the authors have supported their observations using the cell lines with experiments with authentic primary EVT cells, which is essential.

Response 1: We have replaced in the text of the manuscript “EVT cell lines” for “EVT-like cell lines” as requested. As pointed out by the reviewer, all critical experiments were also performed with primary trophoblast EVTs due to concerns associated with the use of cell lines.

Point 2: CoCl2 and hypoxia are not equivalent.  These conditions share features but responses to CoCl2 are not necessarily the equivalent of responses to low oxygen. 

Response 2: We had initially performed the experiments at USU with Swan71 and HTR-8 SVneo cells with CoCl2 and understand that this is not the true equivalent to hypoxia. As we were able to purchase an hypoxic chamber, we repeated the experiments (cell adhesion and migration) in the hypoxic chamber under conditions of 1% Oxygen. The obtained results were similar when we used either CoCl2 or 1% O2. We have added this information in the revised manuscript. As noted in the material and methods section and in the figures, all experiments with primary EVTs were performed in laboratories that had hypoxic chambers set up to 1% oxygen and in the regular setting in the CO2 incubator (21% oxygen).

Point 3:  It would be helpful if the authors provided catalog numbers for the antibodies.

Response 3: Catalog# of Abs have been added in the materials and methods section as requested.

Point 4: Figures - the quality of the images in the manuscript I received was not optimal.  Hopefully, the quality of figures in the published manuscript will be better.

Response 4: Quality of figures should be improved as the resolution of the images submitted should take care of this issue.

Point 5: The authors should be careful regarding their use of terms such as normoxia and hypoxia as applied to cell culture.  Ambient oxygen (~20-21% O2) in an incubator is not normoxia and is not equivalent to the oxygen tension of a cell developing in the placenta/uterus.  Hypoxia is a state defined best by the cellular response to low oxygen.  It might be more advisable to refer to the states as ambient oxygen and low oxygen rather than normoxia and hypoxia.

Response 5: As requested, we have replaced the terms “normoxia” and “hypoxia” for the oxygen concentrations utilized in the experiment.

Point 6: Is the integrin composition of primary EVT cells versus HTR8 or Swan71 cells the same?

Response 6: Primary EVTs, HTR8/SVneo and Swan71 all express a5b1 integrin as reported and as shown in the supplemental figures. The full integrin expression of HTR8/SVneo has been reported in the literature and not much is known about Swan71. It is well-documented that invasive primary EVTs express a5b1, which we have confirmed by IHC and show in the manuscript.

Point 7: Lines 623 and 624, in this sentence HTR8 and Swan71 cell lines are referred to as EVT cell lines.  This is likely not appropriate.  It may be more appropriate to refer to the cell lines as EVT-like cells, as suggested above.  The authors should take a look at the following report:

Lee CQ, Gardner L, Turco M, Zhao N, Murray MJ, Coleman N, Rossant J, Hemberger M, Moffett A. What Is Trophoblast? A Combination of Criteria Define Human First-Trimester Trophoblast. Stem Cell Reports. 2016 Feb 9;6(2):257-72. 

Response 7: As indicated above (in Response 1), we have changed the manner in which we refer to the HTR8/SVneo and Swan71 cells.

Point 8: Are all of the co-authors contributions stated? Stephen Lye?

Response 8: We apologize for the omission and have added the contribution of Dr. Lye to this work.

Reviewer 2 Report

I have reviewed the manuscript entitled “Interaction of pregnancy-specific glycoprotein 1 with integrin α5β1 is a modulator of extravillous trophoblast functions” by Rattila et al.

The authors conducted immunohistochemistry and western blot analysis to demonstrate expression of PSGs by extravillous trophoblast cells. They used PSG1 to demonstrate that EVT cells adhere to PSGs via α5β1 integrins and that one PSG molecule can bind simultaneously to HSPG and can bind to α5β1 integrin thereby connecting EVT with the ECM. Finally the authors observed that the serum concentration of PSGs correlate with PE in African American women when the fetuses were male.

This are several very interesting observations, which merit publication. Nevertheless, there are some points that have to be addressed before publication.

Major comments:

Expression of PSGs by extravillous throphoblast: the authors clearly showed that EVT express PSGs at the protein level by immunohistochemistry. Unfortunately, they did not provide any hint to the amount of expression comparted to the expression of PSG in the syncytiotrophoblast, which was previously detected. A control staining (ideally in the same slide) would be very helpful. In addition, it is not clear, if indeed PSG1 is expressed by the EVT. Although data in Ref. 37 indicated that PSG1 among other PSGs is expressed by iCTB, the consistence of the study would be greatly improved by demonstrating that PSG1 is expressed in the tissue/cells under investigation.

Native PSG1: Since both antibodies used, recognize PSG1, 6, 7, and 8 the purified PSG from the serum may contain all of them – this should be clearly stated in the text. In addition, PSG instead of PSG1 should be used, whenever the possibility exist that other PSGs may be detected. This may be also true for the ELISA results. Please check carefully the whole manuscript for this point.

PSG1 levels in serum: Please comment on the fact that the main part of the paper is dealing with PSGs expressed by EVT, but most likely PSG in the serum is derived from the STB. Do the authors expect that PSG1 secreted by STB contribute to the migration/invasion of EVT by binding to the ECM? If so, I would like to suggest to add an additional fig. to the manuscript showing the hypothesis, how PSGs from different sources favor migration/invasion of EVT.

Adhesion of EVT to PSG1 via α5β1 integrins: What was used as “control protein” for PSG1-His (legend Fig 2B)?

References: There are some flaws in the references, which have to be corrected: Ref. list: 6, 19, 21, 40, 58, 67, Ref. in the text: line 91, 686, and may be more, please check carefully.

Author Response

Point 1: Expression of PSGs by extravillous throphoblast: the authors clearly showed that EVT express PSGs at the protein level by immunohistochemistry. Unfortunately, they did not provide any hint to the amount of expression comparted to the expression of PSG in the syncytiotrophoblast, which was previously detected. A control staining (ideally in the same slide) would be very helpful. In addition, it is not clear, if indeed PSG1 is expressed by the EVT. Although data in Ref. 37 indicated that PSG1 among other PSGs is expressed by iCTB, the consistence of the study would be greatly improved by demonstrating that PSG1 is expressed in the tissue/cells under investigation.

Response 1: As requested by the reviewer, we have added a picture of staining of PSG in the STBs layer alongside staining of the EVTs in the 7-week placenta with anti-PSG Ab#4 in Figure 1. In addition, we have added to the text in the results and discussion sections, information regarding the higher expression of PSGs in the STBs compared to EVTs.

Point 2: Native PSG1: Since both antibodies used, recognize PSG1, 6, 7, and 8 the purified PSG from the serum may contain all of them – this should be clearly stated in the text. In addition, PSG instead of PSG1 should be used, whenever the possibility exist that other PSGs may be detected. This may be also true for the ELISA results. Please check carefully the whole manuscript for this point.

Response 2: Unfortunately, there are no available specific Abs for PSG1 that can be used for IHC at this time but we believe that the references cited in this manuscript, in which PSG1 mRNA was detected in the EVTs by RNA-seq by more than one group, strongly supports the notion that EVTs express PSG1.

Due to the fact that the Abs used for IHC could react with more than one PSG, we have labeled the IHC figure as “PSG” and not “PSG1”. Regarding the native protein, we believe that Ab#4, which was raised by using PSG1 as the immunogen, likely binds to PSG1 with higher affinity than to PSG6, 7 and 8. In any case, to make sure that our native preparation is PSG1, we had submitted the protein for MALDI-TOF MS analysis and the results confirmed that this protein is indeed primarily PSG1 as specific peptides for other PSGs were not detected. We have added this information to the manuscript.

Point 3: PSG1 levels in serum: Please comment on the fact that the main part of the paper is dealing with PSGs expressed by EVT, but most likely PSG in the serum is derived from the STB. Do the authors expect that PSG1 secreted by STB contribute to the migration/invasion of EVT by binding to the ECM? If so, I would like to suggest to add an additional fig. to the manuscript showing the hypothesis, how PSGs from different sources favor migration/invasion of EVT.

Response 3: We thank the Reviewer for raising this important question. We have added the following paragraph in the discussion to address this point “It is likely that a major fraction of PSG1 measured in maternal serum is derived from the multinucleated STB layer of the placenta. Therefore, both the STBs and EVTs can contribute to the circulating levels of PSG1 in the mother. Abnormally low levels of PSG1 in women with preeclampsia may reflect dysfunctional or stressed STBs, which may in turn contribute to the pathogenesis of this placental syndrome (Redman & Staff DOI 1016/j.ajog.2015.08.003). In contrast, in this study we show that staining of PSG in EVT is located at the membrane where its functional interaction with integrin α5 and focal adhesions mediate migration. Due to the limited availability of placental bed biopsies from preeclamptic women we were unable to assess levels of PSG1 in this EVT context in this study. Also whether PSGs affect the initial penetration of the uterine epithelium by blastocytic STBs that occurs in the earliest stages of implantation requires further investigation”.

Point 4: Adhesion of EVT to PSG1 via α5β1 integrins: What was used as “control protein” for PSG1-His (legend Fig 2B)?

Response 4: The control protein utilized in Fig 2B was a recombinant protein produced in our laboratory consisting of the Fc of human IgG1 and this information was added in the revised manuscript. Also as stated in the materials and methods section 2.1, we used different proteins as controls including CEACAM9 (that is also a member of the CEA family as PSGs), or the Fc of human IgG1 with identical results.

Point 5: References: There are some flaws in the references, which have to be corrected: Ref. list: 6, 19, 21, 40, 58, 67, Ref. in the text: line 91, 686, and may be more, please check carefully.

Response 5: We truly apologize for the mistakes in the reference section and thank the reviewer for pointing this out. We have carefully checked and corrected the references in the submitted revised version.

Round 2

Reviewer 2 Report

I have reviewed the revised manuscript entitled “Interaction of pregnancy-specific glycoprotein 1 with integrin α5β1 is a modulator of extravillous trophoblast functions” by Rattila et al.

I have no further concerns.